# Specialized rainforest hunting by *Homo sapiens* ~45,000 years ago

Oshan Wedage[1,2], Noel Amano [1], Michelle C. Langley[3], Katerina Douka[1,4], James Blinkhorn[1,5], Alison Crowther[6], Siran Deraniyagala[7], Nikos Kourampas[8,9], Ian Simpson [8], Nimal Perera[7], Andrea Picin[1], Nicole Boivin[1], Michael Petraglia [1,10] & Patrick Roberts [1]

Defining the distinctive capacities of *Homo sapiens* relative to other hominins is a major focus for human evolutionary studies. It has been argued that the procurement of small, difficult-to-catch, agile prey is a hallmark of complex behavior unique to our species; however, most research in this regard has been limited to the last 20,000 years in Europe and the Levant. Here, we present detailed faunal assemblage and taphonomic data from Fa-Hien Lena Cave in Sri Lanka that demonstrates specialized, sophisticated hunting of semi-arboreal and arboreal monkey and squirrel populations from ca. 45,000 years ago, in a tropical rainforest environment. Facilitated by complex osseous and microlithic technologies, we argue these data highlight that the early capture of small, elusive mammals was part of the plastic behavior of *Homo sapiens* that allowed it to rapidly colonize a series of extreme environments that were apparently untouched by its hominin relatives.

[1] Department of Archaeology, Max Planck Institute for the Science of Human History, 07745 Jena, Germany. [2] Department of History and Archaeology, University of Sri Jayewardenepura, Gangodawila, Nugegoda, Colombo 10250, Sri Lanka. [3] Australian Research Centre for Human Evolution, Environmental Futures Research Institute, Griffith University, Nathan, Brisbane, QLD 4111, Australia. [4] Research Laboratory for Archaeology and the History of Art, University of Oxford, Oxford OX1 3QY, UK. [5] Department of Geography, Royal Holloway, University of London, Egham, Surrey TW20 OEX, UK. [6] Department of Archaeology, University of Queensland, Brisbane, QLD 4072, Australia. [7] Department of Archaeology, Government of Sri Lanka, Colombo 00700, Sri Lanka. [8] Biological & Environmental Science, University of Stirling, Stirling FK9 4LA, UK. [9] Center for Open Learning, University of Edinburgh, Edinburgh EH8 8AQ, UK. [10] Human Origins Program, National Museum of Natural History, Smithsonian Institution, Washington, DC 20013-7012, USA. These authors contributed equally: Oshan Wedage, Noel Amano. Correspondence and requests for materials should be addressed to O.W. (email: wedage@shh.mpg.de) or to N.A. (email: amano@shh.mpg.de) or to M.P. (email: petraglia@shh.mpg.de) or to P.R. (email: roberts@shh.mpg.de)

There is growing evidence that *Homo sapiens* had a unique capacity to adapt to a diversity of extreme environments, both within and beyond Africa, when compared with other members of the genus *Homo*[1]. Nevertheless, studies of migrations of our species into Europe, the Middle East, and Asia have often focused on its increased efficiency in hunting, butchering, and consuming medium-to-large game in open "savanna" settings[2,3]. Alternatively, coastal settings have been highlighted as providing homogeneous, protein-rich resources that stimulated human evolution as well as migration beyond Africa from the Late Pleistocene[4,5]. Focus on these environments has meant that small mammals have been neglected in discussions of the human colonization of new environments, despite the fact that a specialization in their procurement is often considered a feature of technological and behavioral "complexity" or "modernity" unique to our species[6,7]. Concentration on Europe and West Asia in this regard has linked increased usage and capture of agile, but abundant, small mammals to human population growth or climatically driven crises associated with the end of the last glacial period[6]. Nevertheless, the onset and behavioral context of small mammal hunting in other parts of the world, and beyond temperate environments, has remained poorly studied.

From the Late Pleistocene onwards, our species inhabited a number of diverse environments as it dispersed beyond Africa. One of these environments, tropical rainforests, has been considered a barrier to human dispersal[8,9]. This was mainly due to the fact that mammalian megafauna (> 44 kg[10]), thought to have been attractive to Late Pleistocene humans, and even driven to extinction as a result of our species' expansion[11,12], are lacking in these settings[8]. Nevertheless, in Sri Lanka, Southeast Asia, and Melanesia, as well as other parts of the world, the earliest evidence for human occupation is often associated with rainforest environments[13–16]. In Sri Lanka, stable isotope evidence has demonstrated that humans relied on rainforest resources for over 30,000 years[17], perhaps aided by the complex microlith and bone toolkits found at Late Pleistocene and early Holocene sites in the region[15,16,18]. However, owing to a lack of detailed faunal analyses, it has remained unknown which foodstuffs sustained human populations during this time, as well as the hunting strategies employed to obtain them.

We apply new chronometric, taphonomic, archaeozoological, and artifactual analyses to the earliest dated archeological site in Sri Lanka, Fa-Hien Lena (Fig. 1), previously dated to 38,000 years ago[18–20]. Fa-Hien Lena documents the earliest fossil appearance of *H. sapiens* in Sri Lanka, and indeed South Asia, alongside small, quartz microlith technology, and a variety of pointed bone technologies[18–20]. Rainforest mammals, reptiles, molluscs, and plant remains have been identified at the site[18,20,21]. However, human exploitation of specific resources has yet to be directly demonstrated owing to a lack of systematic taphonomic study. Similarly, the early, enigmatic microlith and bone technologies discovered here and elsewhere in South Asia have undergone limited analysis, and their use and adaptive function have remained obscure[19,20,22]. The results of our new multidisciplinary analyses document specialized, sophisticated hunting of semi-arboreal and arboreal prey taxa from ca. 45,000 years ago, in the tropical rainforest environments of Sri Lanka.

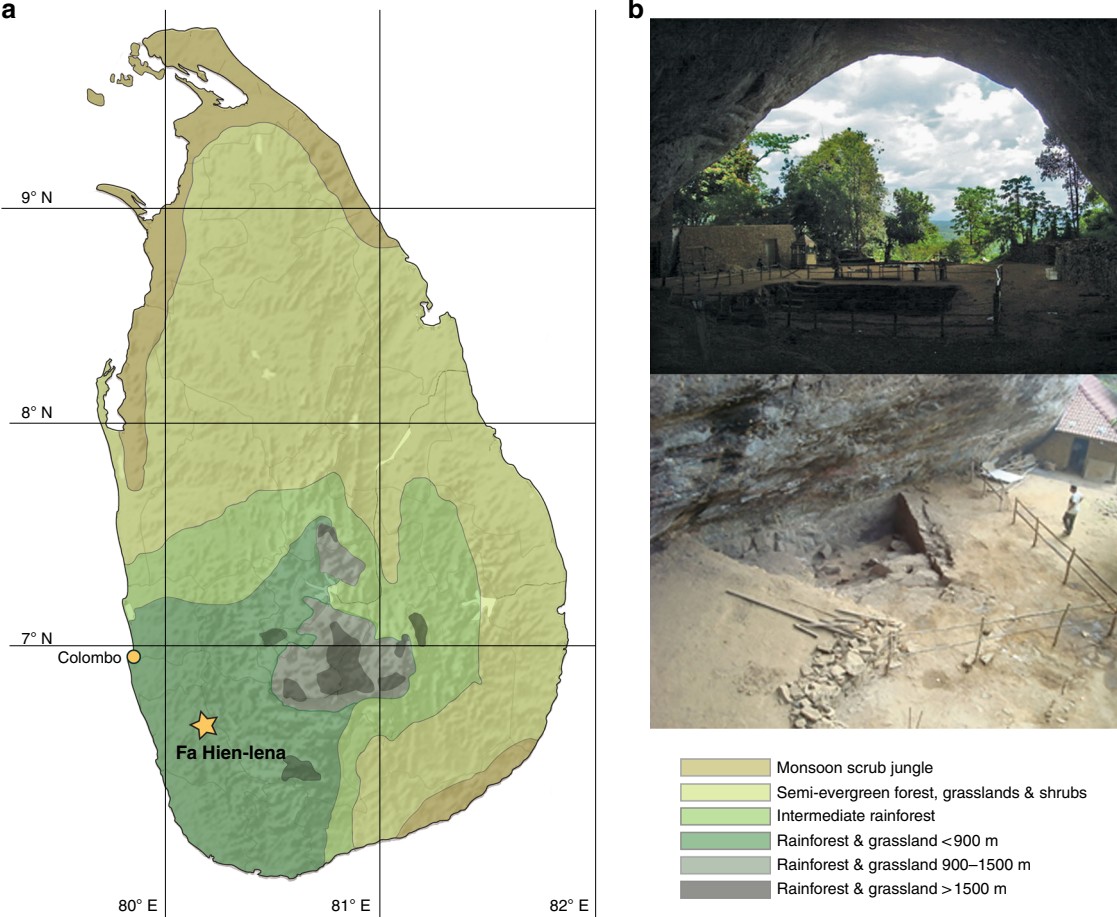

**Fig. 1** Location of Fa-Hien Lena. **a** Map of Sri Lanka showing the location of Fa-Hien Lena and the country's vegetation zones[44,45]. **b** Excavation in Fa-Hien Lena

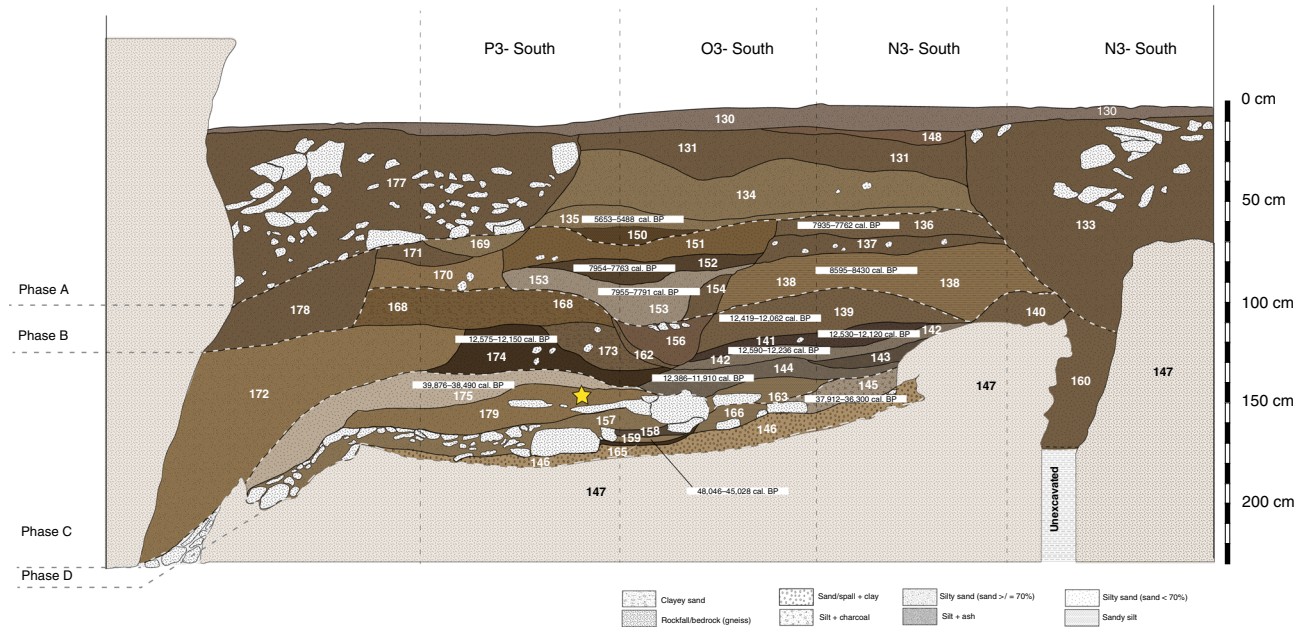

**Fig. 2** Stratigraphy of Fa-Hien Lena. South wall, end of the 2010 excavation. The star indicates the approximate stratigraphic location[19] of the fossils described by Kennedy[23]

## Results

**Stratigraphy and chronology**. We divide the dates available for the site into four distinct phases based on our new, and previous, excavations[19,23]. The phases correspond to concentrations of charcoal, faunal remains, and artifacts, including osseous tools, shell beads, and quartz flakes (Supplementary Figure 1), and represent the major periods of human occupation of the cave. Phase D contains evidence for Late Pleistocene occupation of the cave from *c.* 48,000 to 34,000 cal. BP and probably included several episodes of occupation, each of which may have been relatively short-lived. Phase C spans the Terminal Pleistocene occupation from *c.* 13,000–12,000 cal. BP, while Phases B and A span the Early (8700 to 8000 cal. BP) and Middle (6000 to 4000 cal. BP) Holocene, respectively. One radiocarbon date falls outside these phases (29,120–27,870 cal. BP) and may represent a short-lived episode of human presence in the cave.

These phases also align with major lithostratigraphic changes (Fig. 2). The fill of Fa-Hien Lena consists of *c.* 170 cm of detrital sediment deposited above heavily weathered and karstified gneiss blocks. Phase D consists of pebbly loams and clayey and sandy silt deposits with laminated ash representing intermittent/episodic human occupation and colluvial inwash. The deposits yielded a variety of evidence for human activity, including heavily burned/calcined faunal remains, shell beads, bone tools, and ocher fragments, in addition to micromorphological analyses of sediments evidencing in situ burning (Supplementary Note 1). A roof fall episode appears to have contributed to the exceptional preservation of these deposits by sealing large parts of Phase D from later disturbance. Phase C, which contains the heaviest concentration of artifacts and human occupation debris in the stratigraphy, comprises of a rapidly deposited, heterogeneous mixture of dark colored, organic-rich sandy silty loams. Phases B and A are made up of light colored sandy silts and ash accumulations. For further detailed description, see Supplementary Note 1, Supplementary Tables 1–4, and Supplementary Figures 1–4.

Previous excavation in Fa-Hien-lena produced the oldest human fossils so far in Sri Lanka. Remains of a 5.5–6.5 years old child, mixed with remains of at least two infants as well as a young adult female, were dated based on associated charcoal to 30,600 + 360 BP[23]. These remains were found in layer 4 at the rear of the cave during the 1986 excavations[19] (approximately represented by context 179 during our 2010 excavations) (Fig. 2). Overall, our new data confirm Fa-Hien Lena as the oldest site with *H. sapiens* fossils in Sri Lanka, and wider South Asia[19,20]. They also indicate that Fa-Hien Lena now represents one of the earliest appearances of microlith toolkits and bone tool technocomplexes outside of Africa.

**Zooarchaeology and taphonomy**. Our detailed taphonomic and archaeozoological study of faunal remains at Fa-Hien Lena examined the adaptive context of the first humans on the island. We analyzed a total of 14,485 bone and tooth fragments from the site, 52.6% of which were identified to taxon (number of identified specimens, NISP = 7622). The full dataset can be found in Supplementary Note 2 (see also Supplementary Figures 5–16; Supplementary Tables 5–42). Small mammals (i.e., weighing less than 25 kg) overwhelmingly dominate the faunal assemblage starting from the earliest phase of occupation (*c.* 48,000–34,000 cal. BP). These animals, including carnivores such as the civet cat, account for more than 90% of the NISP, suggesting deliberate hunting from the Late Pleistocene until the Mid-Holocene (Fig. 3) (Supplementary Note 2). Reptiles, including pythons, colubrid snakes, and water monitors, and fish (mostly catfish and carp) are also present in all phases of site occupation. Several of the specimens identified could represent fauna from natural accumulations (e.g., murids and amphibians accumulated by raptors, 1.9% of the total NISP, see Supplementary Note 2). Other specimens, including birds such as swifts and swallows and squamates such as snakes and varanids, could represent the cave's natural faunal communities. However, the high percentage of burning (> 50% in Phase D) in squamate remains suggests that they were most likely utilized by the people that occupied the site (Supplementary Note 2).

There is no significant difference in the distribution of mammals based on body size from the Late Pleistocene to the Mid-Holocene (Supplementary Note 2). Large ungulates, including cervid, suid, and bovid are present throughout the

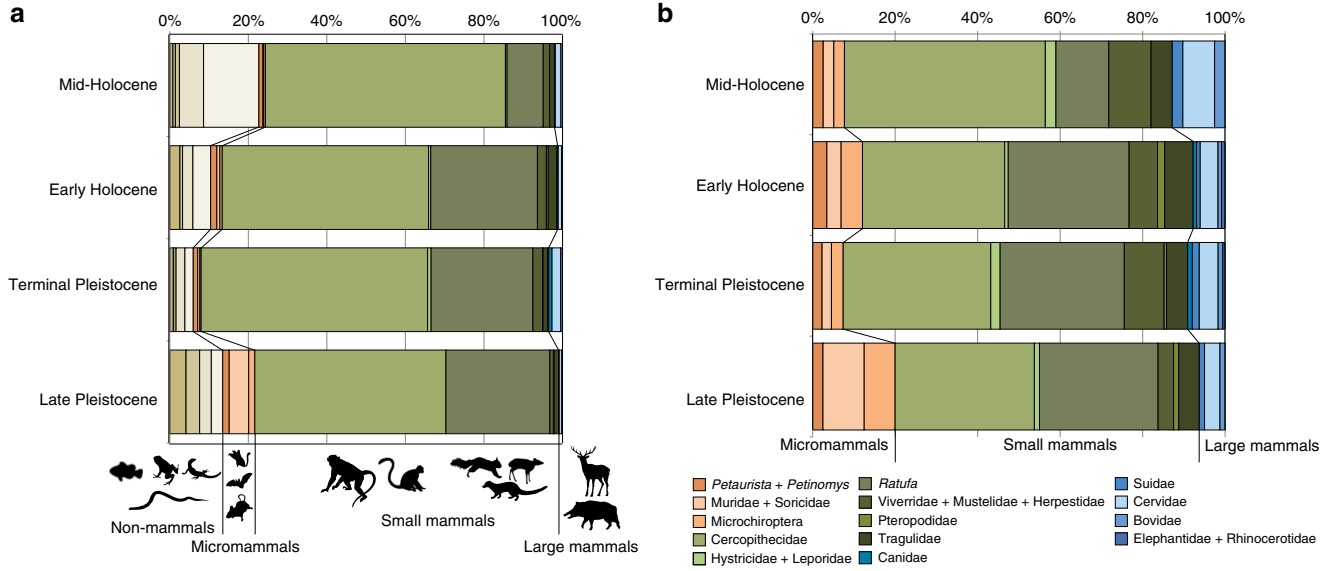

**Fig. 3** Animal taxa identified in Fa-Hien Lena. Distribution of animal taxa identified in the different occupational phases of Fa-Hien Lena based on the number of identified specimens (NISP, **a**) and the minimum number of individuals (MNI, **b**). (Brown: non-mammals; orange: micromammals; green: small mammals; blue: large mammals)

stratigraphy but at very low frequencies (< 4%). Monkeys and tree squirrels overwhelmingly dominate the faunal assemblage in all phases of site occupation, accounting for more than 70% of the identified remains (or 82.4% of the total NISP discounting fauna most likely accumulated by non-human cave dwelling species). They represent 84.7% and 76.3% of the total number of identified specimens in the Terminal and Late Pleistocene layers, respectively (72.3% and 66.7% of the minimum number of individuals, MNI). Of the taxa identified in the Late Pleistocene layers, 48.7% are cercopithecoid monkeys. Deliberate targeting of monkeys continued until the Mid-Holocene, where cercopithecoids represent 61.1% of the number of identified bone and tooth fragments. Three cercopithecoid species are currently present in Sri Lanka: the cercopithecine *Macaca sinica* (toque macaque), the colobine monkeys *Trachypithecus vetulus* (purple-faced langur), and *Semnopithecus priam* (tufted gray langur). These species occur sympatrically and all were identified in the site. Macaques slightly outnumber the leaf monkeys in the faunal assemblage (Supplementary Note 2).

Mortality profiles based on dental eruption and wear suggest that prime-aged adults were deliberately targeted. This, and the fact that the identified monkey species are today mostly arboreal and rarely venture to the ground[24,25], suggests that they were most likely captured by targeted hunting. Trapping usually results in mortality profiles similar to those found in natural populations[26–28]. The presence of bone points and microliths from the outset of site occupation hints at the possible use of projectile technology to hunt arboreal prey (see below, and Supplementary Note 3, Supplementary Figures 17–19, and Supplementary Tables 43 and 44). Modern Southeast Asian hunter-gatherer communities still rely on the use of projectile weapons, including darts and blowpipes, to target arboreal and semi-arboreal taxa[29–31]. The archaeological bone points are consistent in size and breakage patterns with such uses. Regardless of method of capture, entire monkey carcasses were brought and processed in the site as revealed by the pattern of skeletal part abundance (Supplementary Note 2).

Bone fragments with anthropic modification, ranging from burning to butchery marks, were recovered in all phases of site occupation. Butchery marks were recorded on a total of 92 bone fragments (0.64%), the majority of which were from small mammals (92.2%) (Supplementary Note 2). The Late Pleistocene layers yielded a total of nine (0.7% of NISP) bone fragments with clear evidence for butchery, including squirrel, otter, and civet cat long bones from the oldest occupation deposit of the site (Fig. 4). The placement of the cutmarks, both in the shaft surface and the distal epiphyses, is suggestive of a carcass processing sequence that involved disarticulation and defleshing[32]. Burnt and calcined bone fragments represent 19.7% of the total specimens studied (23.9% of the Late Pleistocene assemblage). A high proportion of the small mammal (17.1%) remains identified at the site exhibit evidence for burning, including 16.1% of the monkey remains.

**Bone tool industry**. Primates and giant squirrels appear to have been targeted not just for subsistence, but also for technological production. A total of 36 bone specimens with surface modifications consistent with systematic tool manufacture were recorded in the Late Pleistocene layers (Phase D) of the site (1.3% of the NISP). These consist of 10 fragments of finished implements, including proximally hafted unipoints, mesially hafted bipoints, and small geometric bipoints. The rest are fragments that represent either waste pieces or tool blanks. These specimens are characterized by the presence of heavy surface and/or edge polish and striations or grinding marks. In situ tool production accounts for the high level of fragmentation of cercopithecid bones in all levels of site occupation, but most notably in the Late Pleistocene layers (Supplementary Note 2).

The osseous tools from the Late Pleistocene layers of the site appear to have been manufactured exclusively from cercopithecid long bone fragments, save for one worked macaque canine recorded from the earliest phase of site occupation (context 253) (Fig. 4). Tools and artifacts made from large ungulate bone, teeth, and antler only start to appear during the Terminal Pleistocene. Most of the bone points examined exhibited evidence for damage consistent with high velocity impact (four out of the five points recorded in Phase D, e.g., Fig. 4)[33], which, in addition to what appears to be deliberate targeting of prime age adults, further

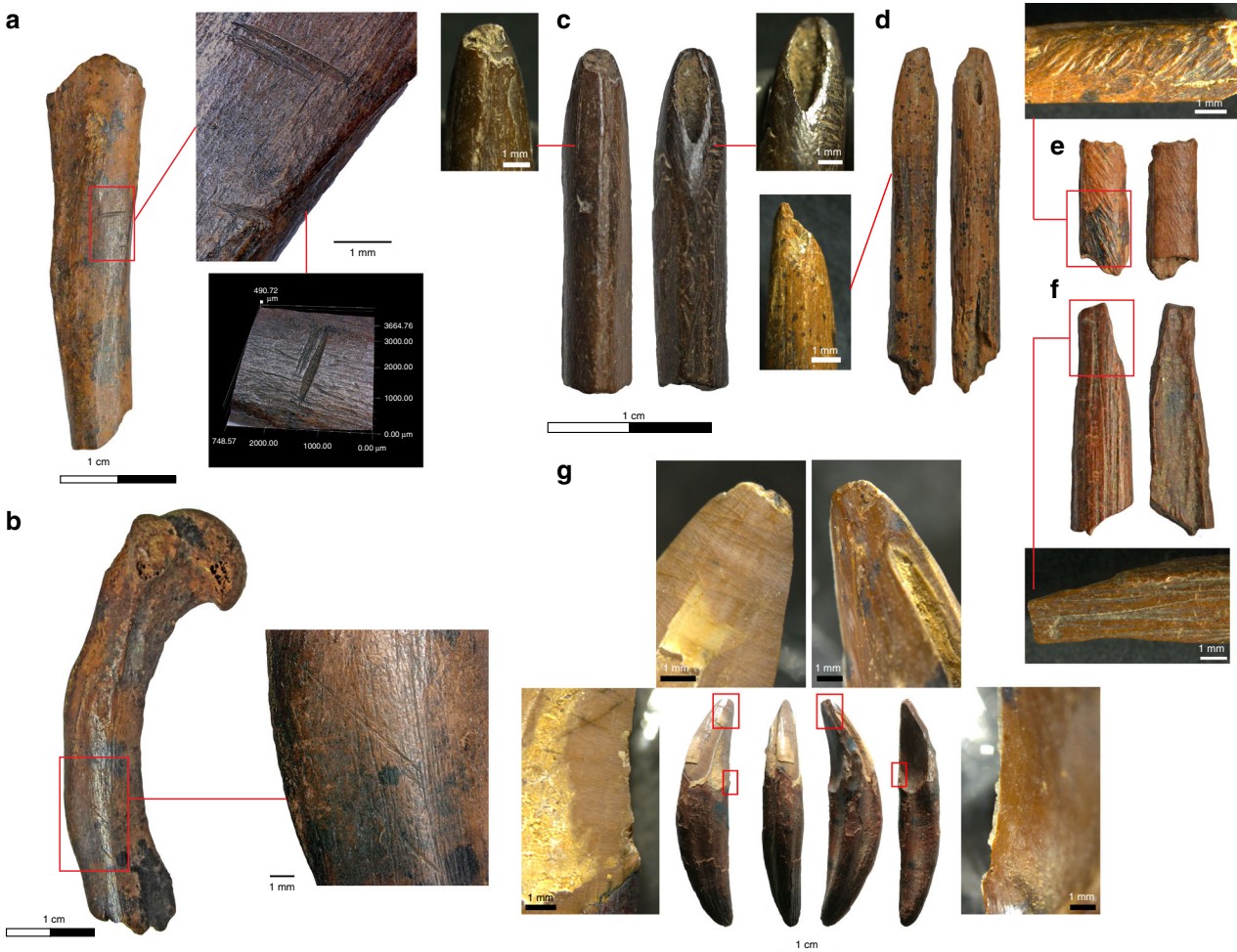

**Fig. 4** Specimens with anthropic modifications from the Late Pleistocene layers of Fa-Hien Lena. Bone fragments with evidence for butchery and osseous tools and artifacts from the earliest phase of occupation at Fa-Hien Lena. **a** Cutmarks on a grizzled giant squirrel (*Ratufa*) tibia. **b** Cutmarks on an otter (*Lutra*) humerus. **c**, **d** Cercopithecid monkey fibula points with evidence for shaping (ground) before high-pressure tip use. **e** Monkey distal fibula shaft fragment with grinding marks. **f** Worked monkey femur shaft fragment. **g** Worked macaque canine showing damage from use in cutting (on sides) and pressure/piercing on tip

suggests that projectile hunting, rather than trapping, was utilized in the exploitation of small semi-arboreal and arboreal game.

## Discussion

The consistently high percentage of arboreal small mammal taxa in all phases of occupation in Fa-Hien Lena is unheard of even among modern ethnographic groups hunting in tropical rain-forest environments[20,34], with perhaps one exception[35]. Even with access to rifle technologies and wire cord and snare traps, modern foragers never show such a bias[34]. Late Pleistocene hunting of arboreal primates has also been demonstrated at the Niah Caves, Borneo from 50,000 to 35,0000 years BP. However, here, in contrast to Fa-Hien Lena, humans seemingly primarily relied on large ungulates as their main source of protein[28]. This, alongside associated evidence from what is, to our knowledge, one of the earliest bone tool assemblages in South and Southeast Asia (contemporaneous with the bone tools from Niah Caves dated between 50,000 and 35,000 years BP[33]), and also one of the earliest beyond Africa, implies highly tuned hunting abilities in tropical rainforest settings upon arrival in this part of the world. Moreover, despite targeting prime age adults, these human populations were able to exploit primate and giant squirrel taxa, often considered to be rainforest game types that are among the

most vulnerable to overhunting[36], from ~45,000 to 4,000 years ago. This suggests close knowledge of life cycles, territories, and sustainable hunting strategies[37].

Discussions of Late Pleistocene dispersals of *H. sapiens* beyond Africa have tended to focus on human reliance on large, mam-malian megafauna that are often linked to open grassland or mixed woodland and grassland environments[8,11]. Alternatively, scholars have focused on reliable coastal resources as providing adaptive corridors for a rapid dispersal around the Indian Ocean, through Southeast Asia, and into Australia[4,5], and also into the Americas[38]. Our current data, demonstrate instead that some of the earliest known *H. sapiens* in Sri Lanka, and indeed in the South Asian tropics, focused on the specialized hunting of small semi-arboreal and arboreal mammals in tropical rainforests. Traditionally, the use of such difficult-to-catch resources has been associated with a "broad spectrum" revolution in the face of growing populations and climate change. However, continued emphasis on prime age animals across over 40,000 years, rather than a broadening of capture strategy, implies no such stress in the tropical rainforests of Sri Lanka. Furthermore, while stable isotopic data from human and faunal tooth enamel in the Wet Zone of Sri Lanka highlights subtle environmental changes from the Late Pleistocene to the Holocene, the persistence of Wet Zone and Intermediate Zone rainforest, as well as human reliance on

this forest[17,39], suggests these environments did not reach carrying capacity.

As a consequence, the utilization of these energetically expensive resources developed in the absence of resource pressure, documents the behavioral and technological flexibility of *H. sapiens*. These complex subsistence strategies appear to be part of the adaptive ecological plasticity of our species that enabled it to inhabit diverse Late Pleistocene habitats across the world[1]. The primary niche of non-*H. sapiens* hominins within and beyond Africa appears to be diverse forest and grassland mosaics in the vicinity of rivers and lakes[40,41]. By contrast, following its evolution in Africa *c.* 300 ka[42], our species came to occupy higher-elevation niches than its hominin predecessors, as well as deserts, palaeoarctic settings, and tropical rainforest habitats stretching across Asia, Melanesia, and North, Central, and South America[1]. Moreover, it was even able to alter and manage environments, such as tropical forests, to meet its own subsistence and cultural needs through ever-intensifying niche-construction[39,43]. Detailed paleoecological and archeological analysis, such as that presented here, offers to yield more insights into the variety of cultural and subsistence strategies that facilitated the eventual colonization of all of the world's continents, and resilience to increasingly extreme Pleistocene climatic fluctuations, that left *H. sapiens* the last hominin standing on the face of the planet.

## Methods

**Chronology and thin section micromorphology**. Fa-Hien Lena (80° 12' 55" E, 6° 38' 55" N) is located in Sri Lanka's Wet Zone region, near the town of Bulath-sinhala, some 75 km southeast of Colombo in a lowland evergreen and semi-evergreen rainforest environment[44,45]. The cave, on the slope of a gneiss cliff, has a *c.* 30 m by 20 m east-facing entrance, an interior that extends *c.* 10 m into the cliff, and two main chambers (termed shelters A and B). First recorded by S.U. Deraniyagala in 1968, Fa-Hien Lena was systematically excavated over several seasons from 1986 to 1988 by W.H. Wijeyapala, and from 2009 to 2012 by a team led by O. Wedage, S.U. Deraniyagala, and N. Perera. Shelter A, the larger of the cave's chambers, was excavated to a depth of over 6 m. However, the shelter's archaeological deposits are disturbed by recent Buddhist constructions. Shelter B, on the other hand, produced a sequence of archaeological deposits spanning from what appears to be the earliest occupation of Sri Lanka by our species (previously dated to 38,000 cal. BP[19,20]) through to the Middle Holocene.

This paper presents the results of the analyses of materials from 2009 to 2012 excavations. We added to existing radiocarbon dates and present a revised stratigraphy for the site. Together with dates previously published[19,23,46,47], which we calibrated using OxCal 4.3[48], we present a total of 30 radiocarbon dates (Supplementary Tables 1–4) that are now available for Fa-Hien Lena, enabling detailed phasing for the site.

A set of undisturbed sediment samples were collected from the excavated profile in clear polyurethane boxes. Sample boxes were labeled, photographed, and plotted on the profile drawing before removal from the profile. Four of these samples, all from Phase D sediments, were selected for micromorphological analysis (Supplementary Figure 2), aiming to understand the depositional history and to access the taphonomic integrity of these earliest occupation deposits. Sample processing, at the Thin Section Micromorphology Laboratory, University of Stirling included air-drying and impregnation with polyester (polylite) resin (http://www.thin.stir.ac.uk/). *c.* 30-μm thick, covered, large format thin sections (7.5 × 11 cm) were manufactured from the hardened impregnated blocks (sample code FH). Thin sections were observed with a polarizing microscope at magnifications of ×12.5 to ×400, using plain polarized (PPL), cross-polarised (XPL), and oblique incident light (OIL). The relative abundance of sediment components was estimated using standard semi-quantitative estimation charts[49,50].

**Zooarchaeology and taphonomy**. We analyzed faunal remains recovered from the 2009 and 2012 excavations of Fa-Hien Lena. All bone fragments from sedimentary contexts with secure radiocarbon dates (including those from deposits sandwiched by dated layers) were included in the analysis. All specimens, including diaphyses and rib shafts, were sorted, counted, and measured (length, width, and thickness) using a digital caliper (Mitutoyo 500–463). Identified specimens were recorded in detail using codes for anatomic zones[51] (e.g., Supplementary Figure 5) that allow the description of bone preservation/fragmentation patterns. Diagnostic dental and skeletal elements were identified to the highest possible taxonomic level using vertebrate comparative collections from the Laboratory of Comparative Anatomy of the *Muséum national d'Histoire naturelle* (MNHN) in Paris and photographs from the mammalian collections of the Field Museum of Natural History and American Museum of Natural History. Following von den Driesch[52], individual

dental specimens and specific anatomical features of diagnostic skeletal elements were measured to differentiate between closely related taxa. The naming of identified taxa follows the nomenclature for mammalian species of Wilson and Reeder[53]. The taxa identified in the sites were assigned to size class based on live weight (modified from refs. [54,55]): (a) micromammals: 100 g to1 kg, (b) small mammals: 1 kg to 25 kg, (c) large mammals class 1: 25 kg to 200 kg, (d) large mammals class 2: 200 kg to 1000 kg, and (e) large mammals class 3: > 1000 kg.

All fragments were examined for natural, animal, and anthropic modifications, including weathering[56,57], abrasion[58], burning, staining, and butchery marks[32]. Bone surface modifications were recorded/observed using an Olympus BX53 light microscope and a Keyence VHX-6000 digital microscope. Burnt bone fragments were identified based on color[59–61]. They were distinguished into different classes based on the degree of burning[56]: (1) partially burnt, (2) charred (blackened), and (3) calcined (partial and complete calcination). Burnt bones were quantified by determining the percentage of the total bone fragments that is comprised of burnt specimens[62,63].

The minimum number of element (MNE) and minimum number of individual (MNI) counts were calculated following a modification of Dobney and Rielly's[51] zonation system. This system is based on the recording of morphologically distinct zones in a skeletal element. The MNE was taken as the total number of non-repeatable zones (i.e., greater than 50% of the diagnostic zone present) for every skeletal element of a taxon. The highest MNE value, considering side and age (epiphyseal fusion and dental wear[64]), was used to estimate the MNI. The MNE counts were converted to minimum animal unit (MAU) values by taking into account the number of times the element occurs in the skeleton. The normed MAU values (% MAU) were used to compare skeletal part representation in the different phases of cave occupation[65].

The length and circumference of long bone fragments were also recorded in relation to complete specimens to measure the extent of bone fragmentation in the assemblage. Long bone fragments were assigned the following fragment circumference and length scores[66,67]: (1) fragments with less than half of the circumference/length of the complete specimen, (2) fragments with more than half of the circumference/length of the complete specimen, and (3) fragments with the complete circumference/length of the complete specimen.

**Artifact analysis**. The technological study of the lithic assemblages of Fa-Hien Lena was carried out following the *chaîne opératoire* concept[68,69], a methodological framework that defines the reconstruction of the various processes of flake production from the procurement of raw materials, through the phases of manufacture and utilization until the final discard. The *chaîne opératoire* concept provides systematic sequences of the flaking activities in which is possible to determine the temporal phase and the position of the artifact produced[70]. The lithic material is composed of 5070 items (Supplementary Tables 43–44). The predominant raw material is quartz, which is abundantly available in the streams nearby.

Few chert blanks were also found and include: a siret flake fragment in context 131 (Phase A); two flakes and a fragment in context 136 (Phase B); a flake fragment in context 141; and one flake and one blade from context 248 (Phase C).

For the analysis of osseous artifacts, the materials were examined using a Zeiss Stemi 508 stereomicroscope fitted with an AxioCam 105 camera. Taphonomic and anthropogenic alterations were identified based on published works[58,71–78] and mapped onto photographs taken with a Canon digital SLR camera.

**Reporting summary**. Further information on research design is available in the Nature Research Reporting Summary linked to this article.

## Data availability

The authors declare that all data supporting the findings of this study are available upon request from the authors. The artifacts and faunal remains from the Fa-Hien Lena excavations are curated at the Department of Archeology, Government of Sri Lanka, under the site code BYP and the suffixes 10, 11, and 12 (denoting the year of excavation). Some materials remained housed at the Max Planck Institute for the Science of Human History to be returned to the Department of Archeology, Government of Sri Lanka by the end of 2019. All of the data reported in the paper are presented in the main text or in the Supplementary Notes, Tables, and Figures.

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

## Acknowledgements
We thank Rev. Chandima Thero, Rev. Anandasagara Thero, and the other monks at the Fa-Hien Lena Cave temple. We would also like to thank Prof. S. Disanayaka, Prof. P.B. Mandawala, Mr. S.A.T.G. Priyantha, Mr L.V.A. De. Mel, and other members at the excavation Branch of the Department of Archeology, Sri Lanka. This research is supported by the research council of the University of Sri Jayewardenepura (RC-USJ) and the Max Planck Institute for the Science of Human History.

## Author contributions
O.W., N.A., M.D.P., and P.R. designed the research; O.W., N.A., M.C.L., K.D., J.B., A.C., S.D., N.K., I.S., A.P., N.B., M.D.P., and P.R. collected the data; O.W., N.A., M.C.L., K.D., J.B., A.C., S.D., N.K., I.S., A.P., N.P., N.B., M.D.P., and P.R. analyzed the data; O.W., N.A., M.C.L., K.D., J.B., A.C., S.D., N.K., I.S., A.P., N.P., N.B., M.D.P., and P.R. wrote the paper.

## Additional information

**Competing interests:** The authors declare no competing interests.

