## [Peer Review File · Nature Communications]

Reviewers' comments:

Reviewer #1 (Remarks to the Author):

The paper is a very good contribution to the developing knowledge of late Pleistocene tropical foragers, a further contribution to the important work on this in recent years in Sri Lanka. It shows that the first (as currently known) *Homo sapiens* foragers here were able to capture small elusive prey like monkeys, probably using projectile technologies of stone and bone. The paper is supported by Supplementary Materials that set out the detailed evidence of stratigraphy, dating, the faunal analysis and stone tool studies. It is clearly written and well evidenced. My only substantive point (under Discussion below) is whether the evidence, whilst really important, is as unique as the paper tries to convey.

Abstract

37,38 'untouched by its hominin relatives' is an assumption given Denisovans, and is a hostage to fortune; rephrase.

Introduction

The Introduction slightly exaggerates the debates as background to the paper's contribution, making them sound as if they are current whereas the references might be two decades or more old: eg 'is' on line 49 should be 'has been' (and grammatically 'have' on 48 should be 'has'), rainforest as a barrier, human dispersals tied to coasts etc.

Stratigraphy and chronology (I'm not sure that these are strictly 'Results')

Ideally Figure S2 ought to be in the main text, as it shows the key stratigraphy and dating.

105 "the oldest site with *H. sapiens* fossils in Sri Lanka": where are the fossils in relation to stratigraphy and chronology?

Zooarchaeology and taphonomy

This is well done, with all the SM to back it up.

Artifact analysis

If the argument is that the first people (Phase D) were probably using bone as well as stone projectile weapons, the argument needs to be clearer in 157-172, rather than first describing worked bone and then making the jump to "they must have made projectiles". I agree with them, but it needs to be better/more carefully argued.

Discussion

178 "the earliest bone tool assemblages in South and Southeast Asia". The earliest bone points in the Niah Caves (Sarawak) are around 50 ka: see Rabett's chapter on the bone tools in the second Niah volume.

As a general point on Niah, the paper ought to be citing relevant material in the 2013 and 2016 monographs rather than the JHE 2007 paper (reference 13) and the Piper and Rabett 2009 paper (reference 28), in the latter case their detailed chapter on the fauna in the 2016 monograph. The latter also shows that the *Homo sapiens* at Niah 50,000 years ago were also able to kill small elusive mammals like monkeys etc even though large mammals (bearded pig) were the primary prey.

185-189 Is a bit repetitive of the Introduction text.

200 'that' should be 'than'

Methods

The micromorphology text reads a bit oddly until you look at the SM as there is no earlier mention of this in the main text. There needs to be a sentence in the Stratigraphy section to flag its contribution. I couldn't find the captions to the main figures.

SM

In Figure S1 there is no explanation of the horizontal axis – is it contexts? I think the SM text on the dates should discuss more about the possible implications for the early dates of the different lab protocols eg Oxford and Beta Analytic.

Reviewer #2 (Remarks to the Author):

This is a very strong submission that warrants publication in a high profile venue such as Nature Communications. The most obvious justification for publication is that it makes an airtight case for initial occupation of the tropical forests of Sri Lanka at 45,000 years ago, pushing back the date of the earliest occupation of this large island by about 7,000 years. This study, then, joins other exciting new work on human dispersals out of Africa (some conducted by members of this same research team) that provides an increasingly fine-grained picture of both the timing and pace of early human dispersal.

But beyond this flashy “first” justification for publication, and of arguable greater importance for understanding the global spread of modern humans, is that it offers a new perspective on the varied strategies humans employed to exploit resources in newly colonized areas. As highlighted in the introduction of the paper, one of the arguments used to buttress the case that modern humans were capable of technological and behavioral complexity not found in other hominins is their ability to capture agile, fast moving small game. The development of this capacity is traditionally maintained to have developed in a context of resource depression caused by either demographic pressure or climate change. Modern humans, with their superior cognitive attributes, it is argued, were able to meet by developing effective strategies for utilizing game species that remained out of reach by other hominins, and so gain competitive advantage in adapting to varied and changing environmental contexts.

This study adds an important new angle on the adaptability of modern humans by convincingly demonstrating how early colonizers of Sri Lanka employed highly specialized and sustainable strategies for exploiting abundant, but difficult to capture, arboreal and semi-arboreal small mammals in the tropical rain forests of their new island home. This study departs from the traditional model for small game use by modern humans in that it sees these strategies as evidence of “adaptive ecological plasticity” of modern humans that does not require climatically or demographically induced resource pressure to be realized. Instead, it is the plasticity of behavior itself that allowed modern humans to occupy niches that remained out of reach of other contemporary or earlier hominin relatives.

This is an important conclusion that I am entirely prepared to accept, having argued that the “broad spectrum revolution” that expanded the diversity and range of resources exploited by humans is a reflection of niche-constructing capacities employed in relatively stable environments in which a range of abundant and predictable resources can be exploited. But I don’t think that the manuscript at present does enough to buttress this conclusion.

Supporting evidence for this conclusion is indeed found in the paper. The duration of the application of these strategies over the various periods of the occupation of Fa-Hien Lena is evidence of their sustainability, which the authors attribute to a “close knowledge of life-cycles and territories” of these game species, is also likely attributable to the low density of human populations over this period that remained well below environmental carrying capacity. The continued emphasis on prime age animals over time would also support the conclusion that these hunting strategies were entirely sustainable and did not destabilize monkey populations, leading to either a steady decrease in the age of individuals exploited or a broadening of the capture strategy to include a fuller range of available animals – young, prime age, and old. There may be other data that might be brought to bear on this argument as well – perhaps paleo-environmental data of plant communities through time, etc.

I think the paper would be strengthened if these important supporting data are marshalled in the discussion to strengthen the case that these strategies developed in the absence of resource pressure, the primary driver in other models of small game utilization by modern humans. And that, instead, they evolved out of the behavioral and technological flexibility of modern humans with cognitive abilities that turbo-charged niche-constructing capabilities that paved the way for their dispersal across the globe. This would not require adding much to the overall length of the paper. Simply a sentence or two to bring these supporting arguments to the fore in making the case should suffice.

On a much more minor note, I think that some discussion of the differentiation between naturally occurring fauna in this cave environment (i.e. the micro-mammals and perhaps some of the reptiles that may represent prey of non-human cave dwelling species) and those purposefully brought into the cave by humans is needed.

I also find the very extensive supplementary information with the detailed discussion and date presentation of the fauna and the lithic studies to be an important, exemplary, addition to the paper.

Response to Reviewers

We wish to thank the Editor and our reviewers for their helpful and constructive comments. We acknowledge and accept the suggestions made by the reviewers 1 and 2 and feel that these have helped to make our paper stronger. In order to highlight our reaction to each and every change we provide a detailed, point-by-point list of responses to the comments made by the Editor and the reviewers below.

Reviewer 1.

The paper is a very good contribution to the developing knowledge of late Pleistocene tropical foragers, a further contribution to the important work on this in recent years in Sri Lanka. It shows that the first (as currently known) Homo sapiens foragers here were able to capture small elusive prey like monkeys, probably using projectile technologies of stone and bone. The paper is supported by Supplementary Materials that set out the detailed evidence of stratigraphy, dating, the faunal analysis and stone tool studies. It is clearly written and well evidenced. My only substantive point (under Discussion below) is whether the evidence, whilst really important, is as unique as the paper tries to convey.

We thank Reviewer 1 for their positive comments in relation to the significance of our study as well as the detail and quality inherent in our Results. Below we outline how we have tried to address their only substantive concern and place the paper more within the context of the research highlighted by the Reviewer.

1. Abstract. 37,38 ‘untouched by its hominin relatives’ is an assumption given Denisovans, and is a hostage to fortune; rephrase.

We have changed this statement to: “that were apparently untouched by its hominin relatives”.

2. Introduction. ‘is’ on line 49 should be ‘has been’ (and grammatically ‘have’ on 48 should be ‘has’), rainforest as a barrier, human dispersals tied to coasts etc.

We have made the suggested changes in the text (line 48-49).

3. Stratigraphy and chronology (I’m not sure that these are strictly ‘Results’).

We think that the new dates obtained as well as the results of reinvestigation of the lithostratigraphy of Fa Hien-lena serve as the basis of the zooarchaeological and taphonomic analyses presented in the paper and should be presented as part of the results. They are the product of new analyses and, as such, are corresponding results to our methods section. We are happy to be guided by the Editor in this regard, however.

The new dates, which have never been published elsewhere, place the human occupation of Sri Lanka c. 7,000 years earlier than results of previous studies. The faunal assemblage we analyzed for this paper was from the reexcavation of Fa Hien-lena from 2009-2012. Therefore, we also

believe it essential to present the stratigraphy of the site to demonstrate the basis of the division of the assemblage into analytical units (i.e. phasing). The results of micromorphological studies, mentioned in line 98-99 and presented in detail in the SI, provide additional evidence of human activity in the lowest layers of site occupation.

4. Ideally Figure S2 ought to be in the main text, as it shows the key stratigraphy and dating. 105 “the oldest site with H. sapiens fossils in Sri Lanka”: where are the fossils in relation to stratigraphy and chronology?

We added the stratigraphy as a figure in the main text (Fig. 2). We also added a short description of the fossils found during the previous excavations (line 104-108) and their location in relation to the presented stratigraphy.

5. If the argument is that the first people (Phase D) were probably using bone as well as stone projectile weapons, the argument needs to be clearer in 157-172, rather than first describing worked bone and then making the jump to “they must have made projectiles”. I agree with them, but it needs to be better/more carefully argued.

We have addressed this by highlighting that most of the projectiles exhibited scarring and fracture patterns consistent with high velocity impact (line 177-185), therefore supporting the idea that these artifacts most likely represent projectile hunting tools. The presence of bone points that were most likely used as armature, in conjunction with the observed mortality profile, we believe strongly supports the idea that the small arboreal mammals were deliberately hunted rather than trapped. Future in-depth analyses of use-wear on lithics recovered from the site is also planned but beyond the scope of this paper.

6. “the earliest bone tool assemblages in South and Southeast Asia”. The earliest bone points in the Niah Caves (Sarawak) are around 50 ka: see Rabett’s chapter on the bone tools in the second Niah volume.

We have added the suggested reference and rephrased the statement as “one of the earliest bone tool assemblages in South and Southeast Asia”. (line 196-197)

7. As a general point on Niah, the paper ought to be citing relevant material in the 2013 and 2016 monographs rather than the JHE 2007 paper (reference 13) and the Piper and Rabett 2009 paper (reference 28), in the latter case their detailed chapter on the fauna in the 2016 monograph. The latter also shows that the Homo sapiens at Niah 50,000 years ago were also able to kill small elusive mammals like monkeys etc even though large mammals (bearded pig) were the primary prey.

We have updated the references and noted the findings of Piper and Rabett in the main text. (line 194-198). We also undertake a more in-depth discussion of the Niah findings in the Introduction and, particularly, the Discussion in general.

8. 185-189 Is a bit repetitive of the Introduction text.

We altered these lines so as to make them less repetitive. However we decided to retain them as we feel that it is necessary to reiterate what we said in the introduction but this time in line with our findings.

9. Line 200 ‘that’ should be ‘than’

We have made the necessary change.

10. The micromorphology text reads a bit oddly until you look at the SM as there is no earlier mention of this in the main text. There needs to be a sentence in the Stratigraphy section to flag its contribution.

We agree with the reviewer and have now added a mention of the results of micromorphological analyses of Phase D sediments in the main text as suggested. (line 90-102).

Reviewer 2.

This is a very strong submission that warrants publication in a high profile venue such as Nature Communications. The most obvious justification for publication is that it makes an airtight case for initial occupation of the tropical forests of Sri Lanka at 45,000 years ago, pushing back the date of the earliest occupation of this large island by about 7,000 years. This study, then, joins other exciting new work on human dispersals out of Africa (some conducted by members of this same research team) that provides an increasingly fine-grained picture of both the timing and pace of early human dispersal.

But beyond this flashy “first” justification for publication, and of arguable greater importance for understanding the global spread of modern humans, is that it offers a new perspective on the varied strategies humans employed to exploit resources in newly colonized areas. As highlighted in the introduction of the paper, one of the arguments used to buttress the case that modern humans were capable of technological and behavioral complexity not found in other hominins is their ability to capture agile, fast moving small game. The development of this capacity is traditionally maintained to have developed in a context of resource depression caused by either demographic pressure or climate change. Modern humans, with their superior cognitive attributes, it is argued, were able to meet by developing effective strategies for utilizing game species that remained out of reach by other hominins, and so gain competitive advantage in adapting to varied and changing environmental contexts.

This study adds an important new angle on the adaptability of modern humans by convincingly demonstrating how early colonizers of Sri Lanka employed highly specialized and sustainable strategies for exploiting abundant, but difficult to capture, arboreal and semi-arboreal small mammals in the tropical rain forests of their new island home. This

study departs from the traditional model for small game use by modern humans in that it sees these strategies as evidence of “adaptive ecological plasticity” of modern humans that does not require climatically or demographically induced resource pressure to be realized. Instead, it is the plasticity of behavior itself that allowed modern humans to occupy niches that remained out of reach of other contemporary or earlier hominin relatives.

This is an important conclusion that I am entirely prepared to accept, having argued that the “broad spectrum revolution” that expanded the diversity and range of resources exploited by humans is a reflection of niche-constructing capacities employed in relatively stable environments in which a range of abundant and predictable resources can be exploited. But I don’t think that the manuscript at present does enough to buttress this conclusion.

Supporting evidence for this conclusion is indeed found in the paper. The duration of the application of these strategies over the various periods of the occupation of Fa-Hien Lena is evidence of their sustainability, which the authors attribute to a “close knowledge of life-cycles and territories” of these game species, is also likely attributable to the low density of human populations over this period that remained well below environmental carrying capacity. The continued emphasis on prime age animals over time would also support the conclusion that these hunting strategies were entirely sustainable and did not destabilize monkey populations, leading to either a steady decrease in the age of individuals exploited or a broadening of the capture strategy to include a fuller range of available animals – young, prime age, and old. There may be other data that might be brought to bear on this argument as well – perhaps paleo-environmental data of plant communities through time, etc.

I think the paper would be strengthened if these important supporting data are marshalled in the discussion to strengthen the case that these strategies developed in the absence of resource pressure, the primary driver in other models of small game utilization by modern humans. And that, instead, they evolved out of the behavioral and technological flexibility of modern humans with cognitive abilities that turbo-charged niche-constructing capabilities that paved the way for their dispersal across the globe. This would not require adding much to the overall length of the paper. Simply a sentence or two to bring these supporting arguments to the fore in making the case should suffice.

We thank Reviewer 2 for their highly positive comments in terms of the interdisciplinary significance of our paper, as well as the detail of our analyses and results. We agree that this is not only a major novel, ‘first’ finding in the context of specialized focused on small arboreal prey but that it also has significant implications for understanding how our species came to utilize these difficult-to-catch and energetically expensive taxa. In a Sri Lankan context this was not seemingly the product of climatic pressures or demographic expansion, as has been

suggested elsewhere, but rather a ‘colonizing’ trait that highlights the plasticity and adaptability of our species as it encountered and specialized in the diversity of environments the world has to offer during the Late Pleistocene. Details in relation to the nature of these adaptations are a hot topic in archaeology and palaeoanthropology at present (see for example the recent article on Tibet), and we are glad that Reviewer 2 sees our paper as contributing to these discussions in a major way.

We agree with their suggestion to highlight the importance of our paper even further in this regard and have made additions to the Discussion as suggested:

Lines 211-223 now reads – *“Traditionally, the use of such difficult-to-catch resources has been associated with a ‘broad spectrum’ revolution in the face of growing populations and climate change. However, continued emphasis on prime age animals across over 40,000 years, rather than a broadening of capture strategy, implies no such stress in the tropical rainforests of Sri Lanka. Furthermore, while stable isotopic data from human and faunal tooth enamel in the Wet Zone of Sri Lanka highlights subtle environmental changes from the Late Pleistocene to the Holocene, the persistence of Wet Zone and Intermediate Zone rainforest, as well as human reliance on this forest^{17,39}, suggests these environments did not reach carrying capacity.*

As a consequence, the utilization of these energetically expensive resources developed in the absence of resource pressure, instead representing the behavioral and technological flexibility of H. sapiens. These complex subsistence strategies appear to be part of the adaptive ecological plasticity of our species that enabled it to inhabit diverse Late Pleistocene habitats across the world¹. Line 227-229 now also reads – “Moreover, it was even able to alter and manage environments, such as tropical forests, to meet its own subsistence and cultural needs through ever-intensifying niche-construction^{39,43}..”

On a much more minor note, I think that some discussion of the differentiation between naturally occurring fauna in this cave environment (i.e. the micro-mammals and perhaps some of the reptiles that may represent prey of non-human cave dwelling species) and those purposefully brought into the cave by humans is needed.

We agree with the reviewer and have now addressed this in lines 125-126. We have now noted the presence of fauna that most likely resulted from natural accumulations as well as those that are known to dwell in caves. In line 137-138 we presented the NISP value discounting specimens most likely accumulated by non-human cave-dwelling species.

However, in line 133-135, we note that although the squamates (varanids and snakes) most likely represent naturally occurring fauna in the cave, the high level of burnt remains suggest that they were also targeted by the foragers that occupied the cave. We hope these extra additions now satisfy Reviewer 2 in this regard.

I also find the very extensive supplementary information with the detailed discussion and date presentation of the fauna and the lithic studies to be an important, exemplary, addition to the paper.

We also thank Reviewer 2 for their positive comments in relation to our Supplementary Information. This paper is the product of over a year of dedicated, detailed analyses and we are delighted it has been recognized as a significant advance in the field.